# A Porous Stone Technique to Measure the Initial Water Uptake by Supplementary Cementitious Materials

**Andras Fehervari \***[ID]**, Will P. Gates, Chathuranga Gallage and Frank Collins**[ID]

Institute for Frontier Materials, Deakin University, Burwood, VIC 3125, Australia;
will.gates@deakin.edu.au (W.P.G.); c.gallage@deakin.edu.au (C.G.); frank.collins@deakin.edu.au (F.C.)
\* Correspondence: andras.fehervari@deakin.edu.au; Tel.: +61-3-9246-8922

**Abstract:** The decades-long use of supplementary cementitious materials (SCMs) as replacements for ordinary Portland cement (OPC) by the cement and concrete industry is undergoing a resurgence in research activities related to goals addressing circular economy activities, as well as reduction in $CO_2$ emissions. Differences in the chemistry, mineralogy and reactivity of SCMs compared to OPC impact the fresh properties of concrete. Some SCMs exhibit greater initial water uptake and thus compete strongly with OPC for water during hydration. This study focuses on the early interaction with water as a primary factor that determines the resulting fresh properties and workability. Currently, no test (standard or otherwise) is available for quantifying initial interactions between water and cementitious materials. A quick and reliable method to measure the initial water uptake of SCMs is presented herein, which relies on their affinity to water. The method enables the calculation of water-to-binder ratios for different SCMs required to achieve the same workability as a reference OPC. The results are then well correlated to measured slump and bleed properties. We propose this simple technique to be used by researchers and industry practitioners to better predict the fresh properties of concretes, mortars, or pastes with SCMs.

**Keywords:** initial water uptake; water affinity; supplementary cementitious material; fresh properties; workability

## 1. Introduction

The enormous quantities of cement used worldwide, amounting to 4 gigatons/year [1], results in contributions of approximately 8% of global anthropogenic $CO_2$ emissions by the cement and concrete industry [2]. The use of supplementary cementitious materials (SCMs) to replace cement in concretes is an effective strategy to reduce carbon emissions and the associated contribution to climate change [1,3–5]. Cement substitution seems to be one of the most effective ways of eliminating $CO_2$ in the process of making cementitious construction materials, because ~60% of the emitted carbon stems from the release of chemically bound $CO_2$ in limestone during cement production [1,3]. Most SCMs used nowadays contain no such structurally bound $CO_2$ in large quantities. While increasing amounts and types of SCMs are used each year [6–8], the carbon footprint of the sector can still be further reduced by improving the substitution rates of ordinary Portland cement (OPC) [1].

At present, SCMs are used for different reasons: to name a few, calcined clay in cementitious blends have been shown to reduce the porosity of mortars and concretes and thus improve some durability properties [9,10]; blends containing good quality fly ash, on the other hand, are often applied where a higher level of workability with the same water-to-binder ratio is required [11]; and limestone and limestone calcined clay cement (LC3) blends are often utilized in applications where the reduction in the carbon footprint is the main priority [3]. However, for all SCM usage, an overarching benefit, and often the driving force, is the lower embodied carbon of the resulting materials.

In this research program, four types of SCMs were studied: calcined clay (i.e., metakaolin), fly ash, limestone and LC3. Calcined clay and limestone were selected because they are available in quantities comparable to the total quantity of OPC used worldwide and thus allow for higher substitution rates and can provide sufficient future supply [12,13]. Furthermore, it has been shown that lower grade clays having a kaolinite content of more than ~50% are adequate as SCM on calcination [13,14]. Finally, fly ash was also studied because many countries have large reserves of such SCMs [15,16].

An unavoidable effect of the addition of SCMs to concrete, mortar or cementitious paste that limits the amount of SCM that can be effectively used is the adverse alteration of the workability of fresh mixes [17]. Fly ash SCM addition, for example, can either increase or decrease water demand when used to replace cement, depending on its quality and carbon content [11]. While not extensively reported in the literature, the use of calcined clay SCMs typically increases the water demand of concretes substantially. LC3 mixtures show similar fresh properties to those of calcined clay; however, even in comprehensive studies (e.g., [18]), fresh behaviour is not discussed in detail. Other studies [14,19] indicate that to obtain a similar slump to that of OPC–only concrete, greater amounts of superplasticizer are needed and even then, a poorer slump retention is observed in LC3 systems. Nair et al. [14] also reported that polycarboxylic ether (PCE)–based admixtures are the most appropriate for calcined clay/LC3 type of cementitious mixes, indicating the sensitivity of calcined clay SCMs to the type of admixture used.

It is often argued whether limestone belongs to the large family of SCMs or simply acts as a filler in cementitious mixtures, despite its involvement in hydration reactions in LC3 type of blends [20]. In binary systems where the cementitious blend consists of cement and limestone only, the latter behaves as an inert filler only [21]. On the other hand, in the presence of, e.g., metakaolin-type aluminosilicates and portlandite, the formation of calcium monocarboaluminate hydrate is energetically favourable [20–22]. The addition of non-heated limestone to cementitious materials is now common practice and has the advantage that it does not result in $CO_2$ liberation and additionally tends to improve the workability of concrete [23,24].

It is generally accepted that various SCMs interact with water in different ways and these interactions affect the observed fresh properties. For example, differences in water absorption and flocculation are parameters deemed to be important in the observed fresh properties of SCM blends [25,26]. The increased water absorption of calcined clay for instance can be explained as a consequence of the calcination step [27], whereby the dehydroxylation process leads to coordinated undersaturated aluminium atoms in the structure of metakaolin which ultimately exhibits a greater rehydration energy and increased solubility [28–30]. The rapid initial rehydration process then results in restricted workability, especially at lower water-to-binder ratios [27]. Fly ash, on the other hand, tends to absorb water via a physical surface interaction and the chemistry of fly ash particles has been shown to have a negligible impact on the water uptake [31]. Thomas (2007) [11] also reported that a 10% fly ash addition should result in at least a 3% reduction in the water demand of a cementitious mix, however, the exact value of water reduction depends on the quality of the fly ash. Finally, similarly to fly ash, limestone has been shown to improve the workability of cementitious mixtures through a surface interaction with water and because of this phenomenon, the fineness of the limestone particles has been found to be key in the workability of limestone-OPC blends [32].

The conventional testing of fresh concrete (e.g., slump, bleed, rheology) indicates the extent to which SCMs can impact the workability of fresh mixes, but most of these tests fail to provide information on the underlying mechanisms that cause changes to fresh properties such as wetting or water adsorption. As such, the application of tests for fresh properties limits progress toward overcoming workability issues. This study addresses this knowledge gap by introducing a simple test method for evaluating the initial interaction of water with cementitious materials—termed the initial water uptake (IWU)—where the

affinity of the cement-SCM blends to water can be quantified and correlated with other fresh properties and cement replacement percentages.

Currently, no standard test exists for measuring the initial water uptake of cementitious materials, despite the known importance of the water-SCM interaction to fresh properties. Up to date studies [33–36] and standards [37,38] deal with the water uptake of hydrated (hardened) cementitious pastes, mortars and concretes only. However, as will be detailed herein, the IWU of various non-hydrated (dry) cementitious materials can differ widely and thus could serve as a useful supplement for the understanding of fresh properties. The proposed method targets applications where SCMs are used, and in circumstances in which the behaviour of a cementitious mix with varying amount of SCMs need to be evaluated in advance (e.g., concrete pumping, form work requirements, placement, compaction, and surface finish). Thus, the IWU may provide fundamental information beneficial in many applications especially in the planning stage when determining the desired mix design. This research also responds to one of the main research requirements of "mastering the workability of fresh concrete" listed in the United Nations' 2017 report on eco-efficient cements and in Scrivener et al. (2018) [3], and thus helps to increase concrete quality, reduce waste, and ultimately saves time and money.

## 2. Materials and Methods

### 2.1. Materials

The main binder and reference material of this study was a general purpose (GP) ordinary Portland cement (OPC) marketed as Eureka GP cement by Independent Cement and Lime Pty Ltd. (Port Melbourne, Victoria, Australia). Calcined clay, fly ash, limestone and LC3 (calcined clay in combination with limestone) were used as supplementary cementitious materials (SCMs). A kaolin pottery clay having >80% kaolinite, was provided by Claypro Australia Pty Ltd. (Bendigo, Victoria, Australia); Blue Circle® fly ash was purchased from Boral Cement (Boral Australia); crushed limestone was obtained from a local garden supply (SoilWorx in Pakenham, Victoria, Australia); and gypsum (for LC3 blends) was sourced from Bunnings Warehouse (Melbourne, Victoria, Australia).

The as-received clay, limestone and gypsum were separately milled in a planetary ball mill l (ND0.4L, Torrey Hills Technologies, LLC, Across International Australia, Melbourne, Vic., Australia) to obtain similar particle size distribution (PSD) to that measured for OPC. The milling program employed was detailed in the Supplementary Data (Table S1). The milled clay material was then calcined in a muffle furnace in our laboratory at 800 °C for 1.5 h following recommendations by Scrivener 2018 [39]. The Fourier transform infrared spectra and powder X-ray diffraction patterns of the as-received and calcined clay material were previously discussed [40], where it was shown that, as anticipated, the kaolinite in the pottery clay transformed to metakaolin following the calcination process applied [41–43].

Calcined clay replaced 5, 10, 15, 20, 30, 45, 60, 75 and 90 wt% of OPC in one series of tests, and fly ash replaced 15 or 30 wt% of the OPC in another. For comparisons, a LC3 mix was made up of 50 wt% clinker, 30 wt% calcined clay, 15 wt% limestone and 5 wt% gypsum blend [12]. A 5 wt% gypsum + 15 wt% limestone + 80 wt% clinker scenario was also studied to understand the effects of limestone (Table 1). Since the composition of OPC was 3 wt% gypsum, 6 wt% limestone and 91 wt% clinker [44,45], the required ratios (10:6:3:1) of clinker, calcined clay, limestone and gypsum were adjusted when mixing LC3; likewise, the levels of gypsum and limestone in the 5 wt% gypsum + 15 wt% limestone + 80 wt% clinker mix were accurately calculated by using the reference OPC as the source of the clinker. When selected, the replacement percentages of various SCMs used in this study (Table 1) were based on the conventionally applied replacement quantities. For example, fly ash–OPC blends usually contain 15–30% fly ash [11,46,47] or LC3 incorporates 30% calcined clay and 15% limestone [12]. For calcined clay, however, very high replacement percentages were applied in order to evaluate the performance of the proposed technique under extremely high SCM contents. These blends were studied to understand how each SCM affected initial water uptake, and thus to understand how IWU impacted fresh properties.

**Table 1.** SCM–OPC blends used in this study.

| Blend | Clinker (%) | Calcined Clay (%) | Fly Ash (%) | Limestone (%) | Gypsum (%) |
|---|---|---|---|---|---|
| 5% calcined clay | 86.45 | 5 | 0 | 5.7 | 2.85 |
| 10% calcined clay | 81.9 | 10 | 0 | 5.4 | 2.7 |
| 15% calcined clay | 77.35 | 15 | 0 | 5.1 | 2.55 |
| 20% calcined clay | 72.8 | 20 | 0 | 4.8 | 2.4 |
| 30% calcined clay | 63.7 | 30 | 0 | 4.2 | 2.1 |
| 45% calcined clay | 50.05 | 45 | 0 | 3.3 | 1.65 |
| 60% calcined clay | 36.4 | 60 | 0 | 2.4 | 1.2 |
| 75% calcined clay | 22.75 | 75 | 0 | 1.5 | 0.75 |
| 90% calcined clay | 9.1 | 90 | 0 | 0.6 | 0.3 |
| 15% fly ash | 77.35 | 0 | 15 | 5.1 | 2.55 |
| 30% fly ash | 63.7 | 0 | 30 | 4.2 | 2.1 |
| LC3 | 50 | 30 | 0 | 15 | 5 |
| 15% limestone | 80 | 0 | 0 | 15 | 5 |

Finally, to investigate the usefulness of the initial water uptake test, in a separate experiment, the water requirements of various SCM–OPC mortar mixes were predicted with the help of the IWU values and their slump values were measured and compared with pre-established values. For these mortar mixes, washed sand fine aggregate with a specific gravity of 2.54 and water absorption of 0.7% was used.

## 2.2. Properties of the Particles

Because of the known relationships that physical properties of particles have on water uptake, a detailed analysis of the morphological features (PSD, cumulative PSD, sphericity, symmetry and aspect ratio) of the OPC and SCMs particles were conducted. A method called Dynamic Digital Image Processing (DDIP, [48]) was followed using an opto-electronic instrument (Camsizer®, Retsch®, Haan, Germany). The instrument takes millions of high-resolution photos of free-flowing particles in random orientations and details statistics for the size, sphericity, symmetry and aspect ratio of the investigated material. A summary of the particle properties determined by the method is provided in Table 2.

The initial water contents of the various cementitious materials were measured with a moisture analyzer (HC103, Mettler Toledo, Melbourne, Vic., Australia). The instrument applies uniform radiant heating to the sample chamber and the resulting weight changes of the sample due to water loss are monitored in real time. The water vapours are allowed to escape through the opening of the sample holder. The instrument auto-controls heating following pre-programmed temperature and sensitivity controls (110 °C drying temperature, 5 g ± 10% start weight and <3 mg/50 s switch-off criterion), and once a constant weight is achieved, the initial moisture content is calculated.

Scanning electron microscopy (SEM) imaging was conducted with a JEOL microscope (JSM-IT300, JEOL Australasia Pty. Ltd., Sydney, NSW, Australia) operating at 5 kV and 20 mA on powder samples deposited on carbon tape-coated sample stages.

**Table 2.** Measured properties of the cementitious particles determined by dynamic digital image processing.

| Parameter | Property |
|---|---|
| x (µm) at $Q_3 = 10\%$ | Particle diameter at which 10% of the particles are smaller, based on volume. |
| X (µm) at $Q_3 = 50\%$ | Particle diameter at which 50% of the particles are smaller, based on volume. |
| X (µm) at $Q_3 = 90\%$ | Particle diameter at which 90% of the particles are smaller, based on volume. |
| SPHT | Sphericity = $4\pi A/P^2$; P—measured perimeter of a particle projection; A—measured area covered by a particle projection; for an ideal sphere, SPHT is expected to be 1. Otherwise, it is smaller than 1. |
| Mean value $SPHT_3$ | Mean value of sphericity, based on volume. |
| Symm | Symmetry = $0.5 \times (1 + \min(r_1/r_2))$; $r_1$ and $r_2$ are distances from the centre of area to the borders in the measuring direction. For asymmetric particles, Symm is <1. "Symm" is the minimum value of the measured set of symmetry values from different directions. |
| Mean value $Symm_3$ | Mean value of symmetry, based on volume. |
| b/l | Aspect ratio = $X_{c\ min}/X_{Fe\ max}$, $X_{c\ min}$—the shortest chord out of the measured set of max. chords $x_c$, $X_{fe\ max}$—the longest Feret diameter out of the measured set of Feret diameters. |
| Mean value $b/l_3$ | Mean value of aspect ratio, based on volume. |
| $Q_0(x)$ | Cumulative distribution, based on a number of particles: number of particles smaller than x in proportion to the total number of particles. |
| $Q_0(x)$ | Density (frequency) distribution, based on the number of particles: first derivative of $Q_0(x)$. |
| $Q_0$–SPHT | Proportion of non-spherical particles, whose sphericity is smaller than a given threshold; based on number of particles. |

### 2.3. Slump and Bleed Tests

To obtain information on the workability of various SCM–OPC pastes and mortars, the standard slump test (Australian Standard 1012.3.1) and a mini slump test were used [49,50]. The mini slump test used a small truncated cone scaled to $D_{top}$ = 19 mm, $D_{bottom}$ = 38 mm and height = 57 mm compared to the standard sized cone ($D_{top}$ = 100 mm, $D_{bottom}$ = 200 mm and height = 300 mm). As for standard sized slump tests, the cone was filled, rodded and levelled with freshly mixed cement paste and the mould was vertically removed to allow the paste to slump. To quantify slump, the area covered by the fresh paste was measured by taking photos incorporating a ruler and processing them with ImageJ software. Thus, rather than reporting the height differential from the top of the cone to the top of the slumped paste, slump is expressed as an area. Five replicates of the mini slump test were performed on each of the tested SCM–OPC blends.

To measure the bleeding properties of the selected SCM–OPC pastes, a "mini bleed test" was used. Note that the mini bleed test used herein is not intended to be a surrogate or alternative test for the standard bleed test (AS 1012.6, 2014, [51]) and it is not a scaled version of the original test such as the mini slump test. Our main intention was to gain some very basic information on the bleeding of various SCM–OPC blends in a small-scale test. Mini bleed tests were run in duplicate. In the mini bleed test, a slightly conical plastic cup ($D_{top}$ = 80 mm, $D_{bottom}$ = 60 mm and height = 100 mm of the filled volume) was filled with ~387 mL freshly mixed cementitious paste (i.e., to obtain a nearly full cup) and the mass of the bleed water collected was measured after 1, 2 and 3 h. Shorter durations were not considered because of the reduced amount of collected bleed water compared to that of a full-size bleed test. The plastic cup was tilted by placing it partly on a 10 mm–thick stage when collecting bleed water and the liquid was collected with a syringe. The amount of bleed water was gravimetrically measured. The bleed container was covered with a lid except during the collection of the bleed water.

For both the scaled mini slump test and non-scaled mini bleed test, a 0.45 water-to-binder ratio was used, considered to be a mix proportion at the higher end of the conventional water-to-binder range for cementitious paste mixes and hence suitable for investigating the early interactions of cementitious particles with water. All paste mixtures

were prepared in a 5 L benchtop mixer following ASTM C305 [52]. Mortar mixes were prepared with various water-to-binder ratios (see Section 3.5) in a 20 L mortar mixer.

### 2.4. Initial Water Uptake Test

The tools and setup for the initial water uptake test described herein are depicted in Figure 1a,b and a detailed schematic with captions of the IWU test setup can be seen in Figure 2. The basis of this proposed test is the Enslin–Neff test traditionally used in geotechnical engineering, particularly in Germany [53–55]. However, the IWU test differs from the Enslin–Neff test by being wholly gravimetric and using a simpler experimental setup. The setup consisted of a triple-layered porous stone stage ($D_{porous\ stone}$ = 100 mm, $W_{porous\ stone}$ = 10 mm) partly immersed in water, a filter paper (Advantec 4A hardened filter paper, $D_{filter\ paper}$ = 90 mm) on top of the upper, dry porous stone and a powder sample spread evenly onto the central part of the filter paper surface. The properties of the filter paper included high wet strength, high pH resistance, slow flow and retention of fine particles (<5 µm). A circular hollow metal ring (Figure 1) $D_{sample} \approx$ 75 mm was used to assist the distribution of the powdered sample on the filter paper (i.e., sample was not allowed within a 7.5 mm "buffer zone" from the edge of the filter paper). The samples were not compacted in any way, just spread with the help of a spatula in a circle with an even thickness (Figure 1); the intention was to mimic the conditions of an actual concrete mix where cement particles are not compacted and have a direct interaction with water. The porous stones served to establish and maintain a uniform water supply.

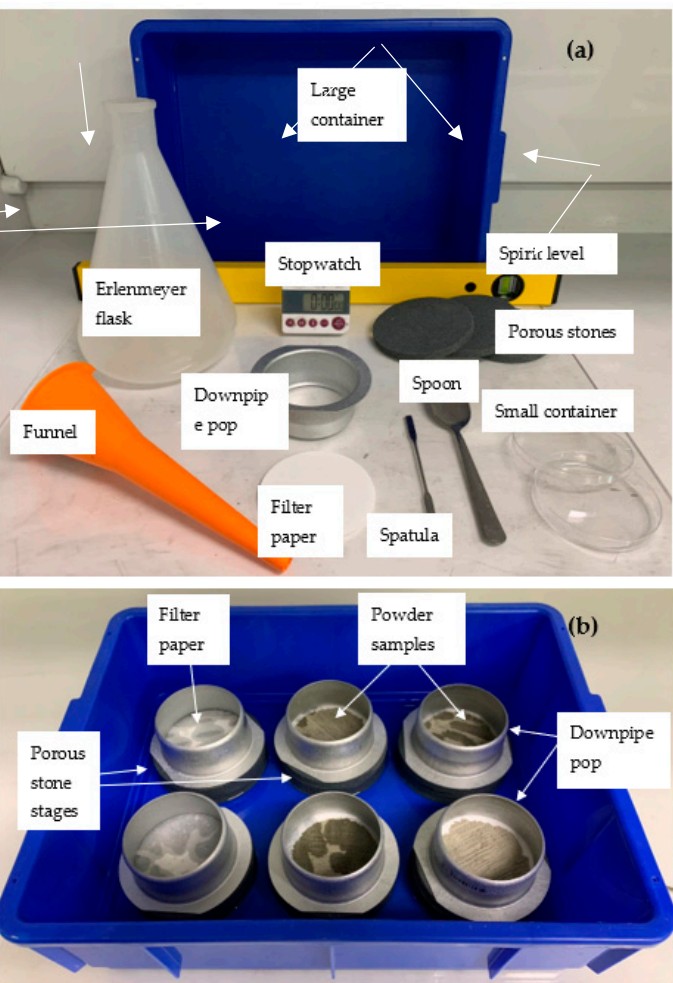

**Figure 1.** View of (**a**) the parts of the initial water uptake test and (**b**) hydrating samples during the test.

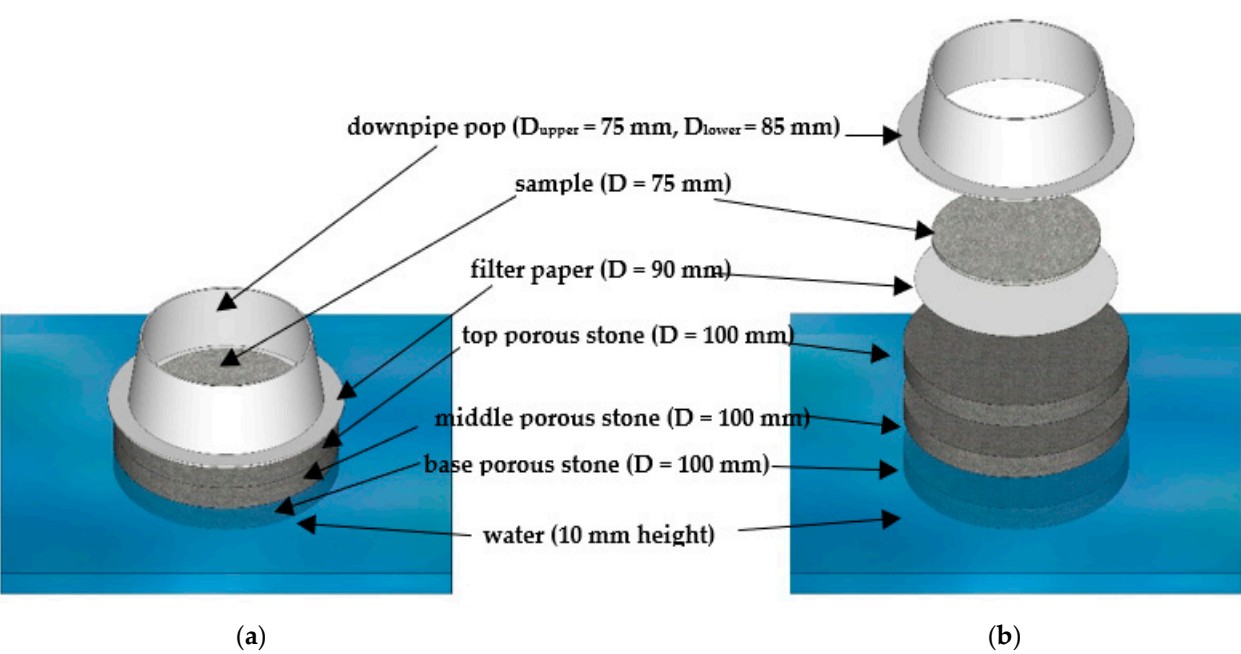

**Figure 2.** Schematic figure of the proposed IWU test: (**a**) side-view; and (**b**) sliced side-view.

In our tests described herein, we stacked three 10 mm–thick porous stones to achieve a suitable capillary height. The triple-layered structure of porous stones slowed the rapid capillary action of water and also provided flexibility in the test by enabling us to use the top porous stone (in its dry state) with the filter paper and initially dry sample as a convenient platform for separate weighing. Although not trialled, we considered a single thick porous stone of sufficient thickness to be fully interchangeable with the proposed triple-layered porous stone stage.

A detailed flowchart of the IWU measurement is depicted in Figure 3. The porous stone stages were placed in a large plastic container (6 of them fit into one container, Figure 1b) and the samples were hydrated through the porous stones by capillary action. A spirit level was used to ensure the assembly was uniformly level. During the IWU test, the container was filled with tap water (in our setup, 985 g $\pm$ 0.3% was used) level to the top of the bottom porous stone (i.e., 10 mm height) and 5 g $\pm$ 0.05 g samples, pre-weighed on the tared filter paper and top porous stone, were allowed to hydrate for 1 min, after which the wet sample (+wet filter paper) was weighed (see below). The height difference between the surface of the water and the bottom of the sample was in all cases 20.15 $\pm$ 0.05 mm (i.e., the thickness of two porous stones and the filter paper). Information about the development of the proposed method and the justification of the test parameters are discussed in Section 3.6 in greater detail.

To ensure good connection between the filter paper and the porous stone, the hollow metal ring with its rimmed side (D = 85 mm) (Figures 1b and 3) was placed around the edges of the filter paper and enclosing the sample. We used a common and inexpensive plumbing accessory (downpipe pop) with d = 75 mm at the non-rimmed side and 85 mm diameter at the rimmed side, for which the difference in diameter and the combination of having rimmed and non-rimmed ends made it suitable for the given application. The sharp-edged 75 mm side was used for uniformly spreading the sample within a 75 mm-diameter circle and the 85 mm rimmed side was used to hold the 90 mm filter paper in place during the test and eliminate any air bubble under the filter paper due to inappropriate connection between the filter paper and the top porous stone. The selected downpipe pop can be purchased in most hardware stores.

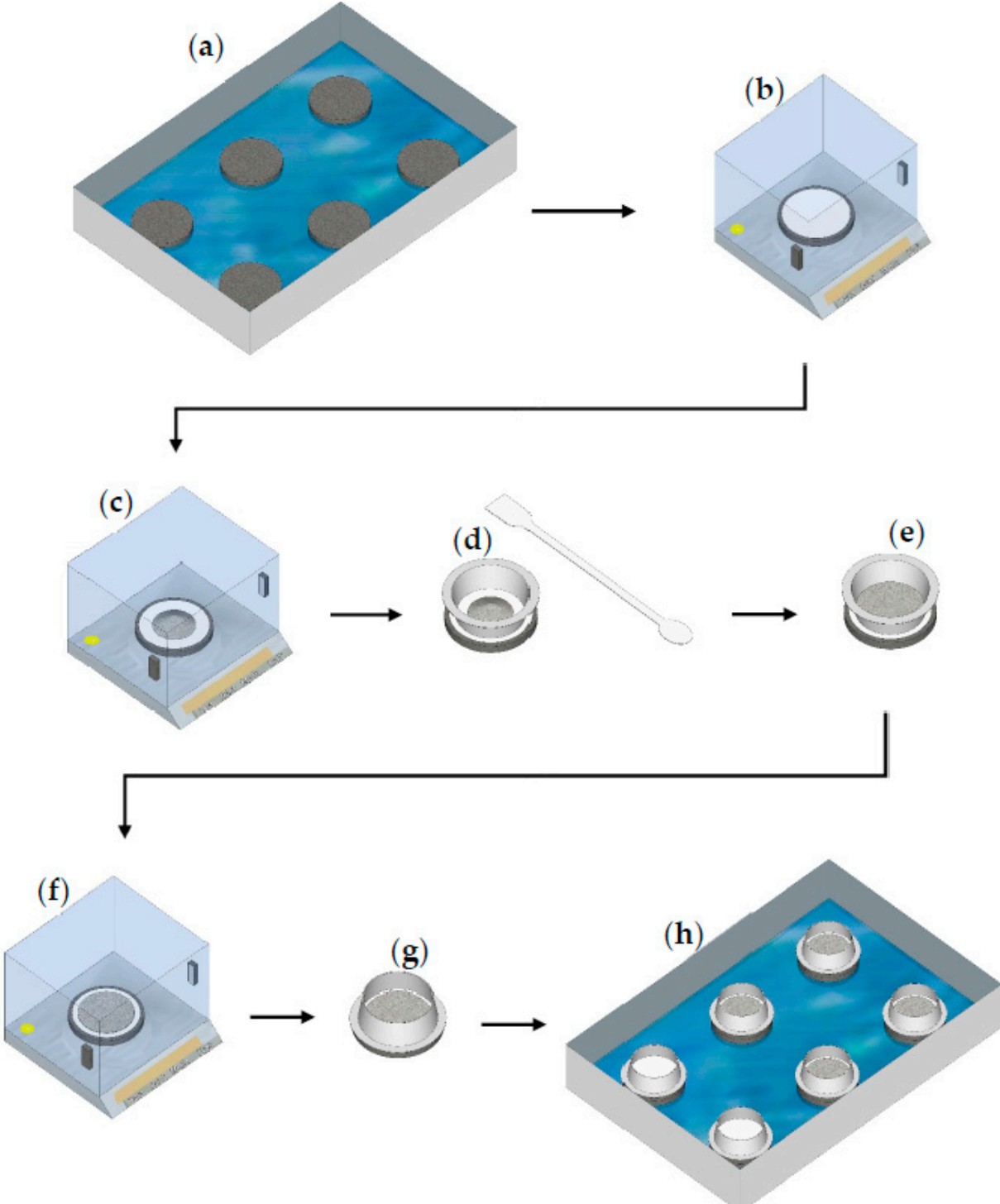

**Figure 3.** Steps of the initial water uptake test: (**a**) place two porous stones on top of each other in a large plastic container with water (water level = height of one porous stone); (**b**) tare the mass of the third (dry) porous stone + filter paper; (**c**) measure out ~5 g of sample; (**d**,**e**) with the help of a spatula and a downpipe pop, create a thin layer of sample with a 75 mm diameter; (**f**) re-weigh the sample and record the value; (**g**) place the downpipe pop around the flattened sample with the rimmed side down (D = 85 mm); and (**h**) start the IWU test by placing the third porous stone with the sample on top of the porous stone stack (containing two porous stones) in the large container with water (run the test for 1 min).

Following the 1 min hydration period, the filter papers with the wet samples were removed using a spatula and placed in pre-weighed and labelled flat plastic containers with lids to minimize evaporation. The masses of the wet filter papers + wet samples

were measured immediately after collection. The collection of all six samples took 1 min (10 s/sample). Obtaining replicates over this time difference was dealt with by incrementally delaying the start-of-wetting of samples. Initially, two porous stones (on top of each other) were placed in the plastic container (at six different locations) and water was added to the container to the level of the top of the bottom porous stones. The top porous stones, together with the dry pre-weighed filter paper and samples, were placed on top of the double-layered porous stones in intervals of 10 s. Sample collection then was conducted in the correct order and thus all samples were hydrated for 60 s ± 1 s. Note that samples were hydrated with water migrating upwards due to capillarity through the porous stones and water reached the filter paper and sample within a few seconds.

For non-experienced users, it is recommended to initially conduct the 1 min test by hydrating and separately weighing each sample rather than conducting the process sequentially (i.e., 10 s apart) as described. We compared the two different methods and measured a ~0.4% ($AVG_{separate}$ = 47.93% vs. $AVG_{sequential}$ = 48.36%) difference in the measured average water contents of OPC after one-minute hydration, although the individual method showed a slightly greater standard deviation compared to the sequential method (1.91 vs. 1.02). Nonetheless, the tests showed that the small differences in the hydraulic height differential had a negligible impact on the measured initial water uptake (the "one at a time" method had initially a slightly greater water head because the water migrated only into one sample at the time) and verified that the driving force for water uptake was the affinity to water of the tested cementitious material.

The aim of the proposed experimental setup was to quantify the initial affinity of various cementitious materials to water; thus, the 1 min duration was selected because we desired to focus on the initial interactions between water and various cementitious materials. However, shorter and longer durations and different sample sizes were also trialled—the results of which are reported in the "Results and Discussion". In our experience, a 5 g powdered sample with 1 min hydration time was found to be the most appropriate test setup for quantifying meaningful differences in the IWU of cementitious materials.

In our test program, each SCM–OPC blend was tested in four replicates and 12 porous stages were used (one measurement required three stages) with the following arrangements: (1) One large plastic container contained 6 stages; (2) two containers were used for each test (6 × 2 = 12 stages); (3) each container contained two stages with filter paper only, two stages with 5 g OPC as reference material and two stages with 5 g of the tested SCM–OPC blend. Ultimately, the amount of water taken up by the SCM–OPC blend was related to the amount taken up by the reference OPC, and the filter paper references were used to estimate how much water was held by the filter paper alone under the given laboratory conditions. With systematic testing, we observed that variations in relative humidity, ambient air temperature and water temperature all influenced the initial water uptake of the blends, reference OPC and the filter paper (see Section 3.5). Hence, the testing of empty "blank" filter papers and determining the ratio of the water uptake of the SCM–OPC blend relative to the water uptake of the reference OPC was established to control these experimental errors. To calculate the IWU ratio, first the gravimetric amount of water ($m_{water,sample+FP}$) taken up by 5 g of cementitious material (either reference OPC or SCM–OPC blend) and filter paper was determined according to Equation (1):

$$m_{water,sample+FP} = m_{wet,final} - m_{dry,FP} - m_{dry,sample} - m_{container} \qquad (1)$$

where $m_{wet,final}$ was the mass of the container used for collecting the wet samples together with the wet filter paper and wet sample (if any) at the end of the test, $m_{dry,FP}$ was the initial mass of the dry filter paper, $m_{dry,sample}$ was the original mass of the dry sample before hydration (5 g ± 0.05 g, initial water content 0.2–0.9%), and $m_{container}$ was the mass of the

plastic container used for collecting the wet samples. Then, the mass of water absorbed by the filter paper ($m_{water,FP}$) was established according to Equation (2):

$$m_{water,FP} = m_{wet,final,FP} - m_{dry,FP} - m_{container} \tag{2}$$

where $m_{wet,final,FP}$ is the mass of the wet reference filter paper together with the mass of container in which it was placed after the test. The gravimetric water content ($GWC_{sample}$) of the sample (OPC or SCM–OPC blend) was then determined by the difference between Equation (1), 2 on averaged samples and filter papers:

$$GWC_{sample} = \{[m_{water,sample+FP} - AVG(m_{water,FP})]/m_{dry,sample}\} \tag{3}$$

where $AVG(m_{water,FP})$ is the average amount of water in the filter paper after 1 min hydration (it is the average of four measurements). Finally, the IWU ratio was calculated by gathering the water contents of the reference OPC and the tested SCM–OPC blend in increasing orders (starting with the smallest water content value measured for the given sample) and taking the ratio of the smallest values ($GWC_{OPC-SCM,min}$ and $GWC_{OPC,min}$), second smallest values ($GWC_{OPC-SCM,2}$ and $GWC_{OPC,2}$), third smallest values ($GWC_{OPC-SCM,3}/GWC_{OPC,3}$) and largest values ($GWC_{OPC-SCM,max}$ and $GWC_{OPC,max}$) and averaging the so obtained ratios:

$$IWU = AVG\ (Ratio_{1-4}) \tag{4}$$

$$Ratio_1 = GWC_{OPC-SCM,min}/GWC_{OPC,min} \tag{5}$$

$$Ratio_2 = GWC_{OPC-SCM,2}/GWC_{OPC,2} \tag{6}$$

$$Ratio_3 = GWC_{OPC-SCM,3}/GWC_{OPC,3} \tag{7}$$

$$Ratio_4 = GWC_{OPC-SCM,max}/GWC_{OPC,max} \tag{8}$$

An IWU ratio higher than 1 indicated a given SCM–OPC blend's greater affinity to water compared to that of OPC and conversely a ratio lower than 1 was observed for blends with lower water uptake relative to the reference OPC.

### 3. Results and Discussion

*3.1. Particle Size, Shape and Initial Water Content*

The physical properties of the tested cementitious materials were considered important when investigating their initial interactions with water and hence the particle size distribution, symmetry, aspect ratio and sphericity properties of OPC and milled calcined clay, limestone and gypsum were studied with dynamic digital image processing (Table 3). On a volumetric basis, all investigated SCMs contained particles smaller than ~33 μm at Q3 = 90% and 90% of the OPC particles were smaller than 36 μm. Limestone and gypsum samples were observed to be slightly finer than the OPC and calcined clay (second and third rows in Table 3), as 50% of the limestone and gypsum particles (based on volume) were smaller than 6.5 μm and 11.4 μm, respectively, whereas for the other two cementitious materials $x_{OPC}$ = 17.3 μm and $x_{calcined\ clay}$ = 14.5 μm at $Q_3$ = 50%. It seemed that limestone and gypsum contained more of the very fine fractions as well (first line in Table 3).

The shapes of the investigated SCMs and OPC were not greatly different. The aspect ratio and symmetry of all tested particles were very similar; however, the milled limestone and gypsum particles were probably slightly more spherical compared with the OPC and milled calcined clay particles. Since OPC and calcined clay were studied in detail in this research program, their properties were further investigated on the basis of particle numbers rather than simply based on volume (Figure 4a,b).

**Table 3.** Properties of the selected SCMs and OPC.

| Parameter | OPC | Calcined Clay | Limestone | Gypsum |
|---|---|---|---|---|
| $x$ ($\mu$m) at $Q_3$ = 10% | 4.9 | 4.4 | 0.6 | 1.4 |
| $x$ ($\mu$m) at $Q_3$ = 50% | 17.3 | 14.5 | 6.5 | 11.4 |
| $x$ ($\mu$m) at $Q_3$ = 90% | 36.0 | 33.0 | 28.9 | 31.3 |
| Mean value $SPHT_3$ | 0.7967 | 0.8199 | 0.8455 | 0.8452 |
| Mean value $Symm_3$ | 0.8998 | 0.9074 | 0.9044 | 0.9110 |
| Mean value $b/l_3$ | 0.7729 | 0.7815 | 0.7694 | 0.7743 |

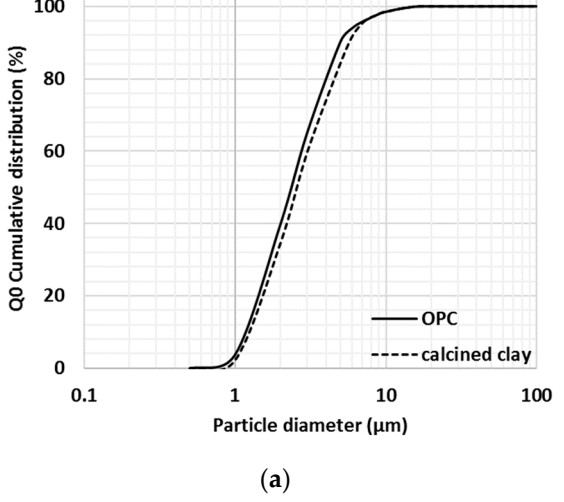

(**a**)

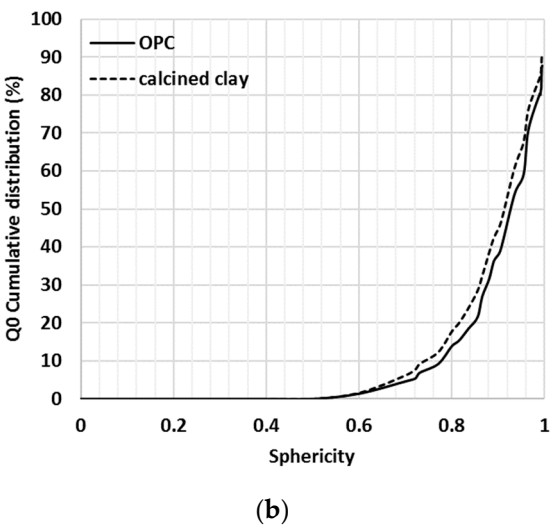

(**b**)

**Figure 4.** Particle size (**a**) and sphericity (**b**) of calcined clay and cement particles (re-plotted from Fehervari et al., 2019 [40]).

According to the obtained particle size distribution (Figure 4a) and sphericity properties (Figure 4b), the milled calcined clay particles were similar in size and shape to that of reference cement particles. Based on a number of particles, 90% of the particles were smaller than 50 $\mu$m and only ~15% of the OPC and calcined clay particles had sphericity lower than 0.8 (one being perfectly spherical).

In summary, the tested materials contained mainly very fine, symmetric and high-sphericity particles. Differences in shape and size amongst the tested OPC and SCM particles were minimal and thus their effects on initial water uptake can be considered negligible. However, the properties of the fly ash particles were not studied with DDIP, and we used fly ash in some of the experiments. Fly ash particles are known to be highly spherical [17,56,57], and according to the data sheet of the manufacturer, 87% of fly ash particles passed the 45-micron sieve.

The average initial water contents of OPC, calcined clay, fly ash and limestone were 0.74%, 0.26%, 0.34% and 0.57%, respectively, and showed negligible variations based on three measurements on three different days. When calculating the initial water uptake ratios, the initial water contents were incorporated in the equation (Equation (1)).

*3.2. Mini Slump*

The mini slump test indicated clearly opposite trends from addition of fly ash compared to calcined clay. In general, the measured area of the mini slump test decreased with the amount of calcined clay used and increased with the incorporation of fly ash (Figure 5). These findings reinforce that, as discussed in the Introduction, the properties of calcined clay impact initial water uptake, and underpin the implications for workability when a larger portion of calcined clayed is blended together with OPC. If only limestone was used in the blend, then a higher workability was obtained and a correspondingly greater mini

slump value was measured. These results suggest that the incorporation of fly ash and limestone tend to improve workability, whereas the addition of the calcined clay SCM is detrimental to the flow properties of the freshly mixed cementitious pastes.

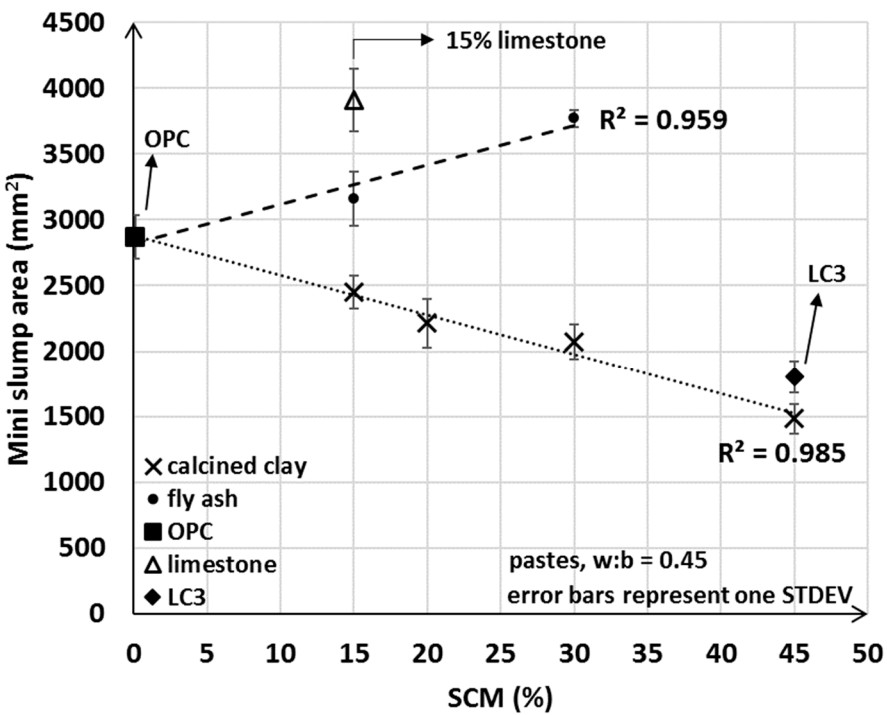

**Figure 5.** Average mini slump values of OPC, limestone, LC3 and various fly ash and calcined clay blends.

Although it was anticipated that the mini slump areas for LC3 pastes would be greater than that for the 30% calcined clay only blend (LC3 incorporated 15% limestone in addition to the 30% calcined clay), our results showed that LC3 had lower workability compared to the 30% calcined clay blend. The mini slump (and slump) test provides essential information on the flow properties (i.e., static yield stress [58]) of a fresh mix. However, the test does not explain possible causes for reduced workability upon calcined clay addition, or the unexpectedly lower workability of LC3. As will be discussed in detail in Section 3.4, the initial water uptake test developed herein fills this gap in providing fundamental information on the water–cementitious material interaction and the observed trends in the measured min slump values.

### 3.3. Mini Bleed Test

The non-cumulative bleed values of various SCM–OPC blends are depicted in Figure 6. Since the main aim of this study was to investigate the initial interactions between the cementitious materials and water, the focus was on comparing the bleed quantities from the various blends at the shortest selected bleed time (i.e., one hour) (Figure 7).

Bleed values upon the addition of calcined clay decreased with the amount used, showing a strong linear correlation ($R^2 = 0.993$) between the amount of bleed water and the percentage of calcined clay. A 15% fly ash addition increased the amount of bleed water collected by 29%; however, replacement of 30% OPC with fly ash showed comparable bleed to that of OPC alone. Limestone slightly increased and the combination of limestone and calcined clay (LC3) decreased the bleeding of the tested cement pastes. For LC3 mixes, no bleed water was detected within the first 3 h of bleeding. These results indicate enhanced interactions between water and calcined clay containing blends and weaker interaction for other types of SCMs studied. Trends observed herein will be explained in light of the initial water uptake in the next Section 3.4.

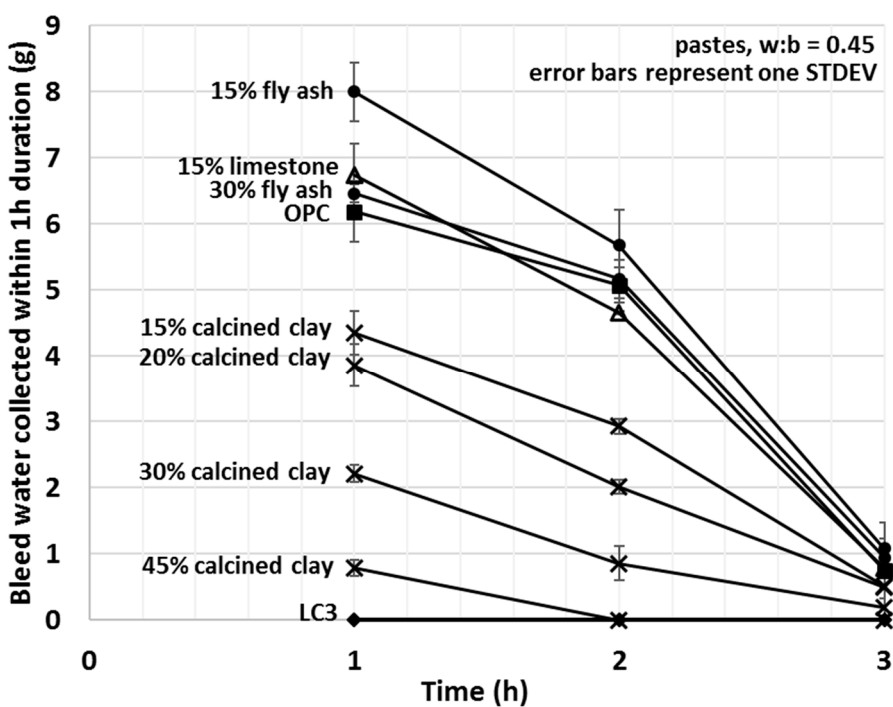

**Figure 6.** Average mini bleed values of OPC, limestone, LC3 and various fly ash and calcined clay blends after one, two and three hours (each data point corresponds to a 60 min bleeding period, the curves are non-cumulative).

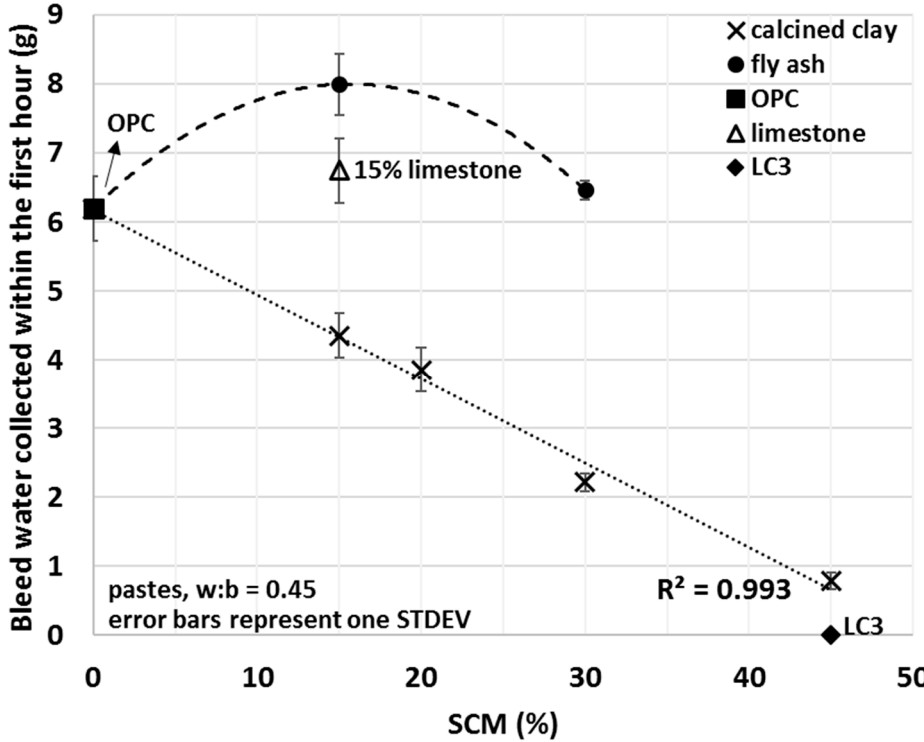

**Figure 7.** Average mini bleed values of OPC, limestone, LC3 and various fly ash and calcined clay blends collected within the first hour of bleeding.

### 3.4. Initial Water Uptake

The IWU ratio of calcined clay–OPC blends was measured on an extended range of OPC replacement with up to 90% SCM content (Figure 8). Results indicated that calcined clay–OPC blends with greater calcined clay content took up more water. However, 60–90% OPC replacement with calcined clay is generally considered to be a non-viable option in most cases, and the intention was to study the effect of calcined clay on the initial water uptake of SCM blends to gain a deeper understanding of how the proposed IWU method performs under extreme conditions (e.g., when SCMs constitute the major component of the blend). Based on seven different calcined clay percentages (15–90%), a strongly linear correlation ($R^2 = 0.992$) was found between the IWU ratio and the amount of clay blended together with OPC (Figure 8); however, the same linearity did not seem to exist at lower calcined clay percentages (0–10% calcined clay). In other words, the proposed IWU test cannot properly differentiate the IWU ratios for ≤10% calcined clay SCM.

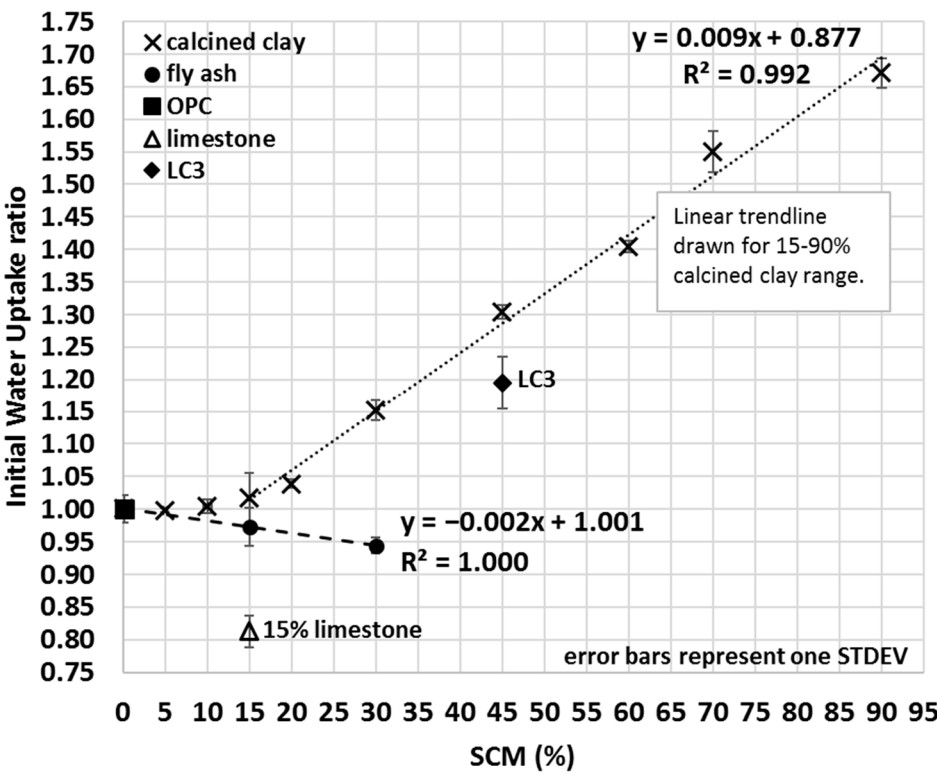

**Figure 8.** Average initial water uptake ratios of OPC, limestone, LC3 and various fly ash and calcined clay blends.

The sensitivity of the test can be increased by increasing the capillarity height differential (distance between the top of the water to the bottom of the sample), for example, by using less water (or using a thicker porous stone). This increased capillarity differential will enhance the relative impact of the sample itself on the measured water uptake ratio because the larger distance for water to migrate diminishes the capillary force at the bottom of the sample, and thus samples with lower matric suction will take up less water. Samples with higher affinity for water will exert greater suction and draw water to greater capillary height differential. In Figure 9 (● symbols), for instance, the height difference between the water surface and sample was increased from 20.15 ± 0.05 mm to 23.48 ± 0.05 mm by using less water (V = 656 g ± 0.3% instead of 985 g ± 0.3%), and consequently, greater IWU ratios of the calcined clay blends were measure compared to that of the original IWU test (X symbols in Figure 9). Thus, although the total amount of water that migrated into the samples was less with a greater capillary height differential, the water uptake by a sample with greater matric suction was enhanced relative to one with lower matric suction. More

importantly, by increasing the capillary height differential by only a few mm, a strongly linear correlation between the amount of calcined clay used in the blend and the measured IWU ratio was observed, having a greater slope (i.e., greater sensitivity) compared to that of the original method. Modifications similar to this can be easily achieved and demonstrate the ability of the IWU test to be custom-made for various types and amounts of SCMs. While the original IWU test (with the height difference of 20.15 ± 0.05 mm) provided a strong linear relationship between the IWU ratio and percentage of SCM used at or above 15% of calcined clay replacement, this could be used to optimize lower replacement levels. This information is highly useful in explaining the measured slump or bleed of the various calcined-clay blends or to evaluate the possible pathways of hydration reactions that take place as the material cures.

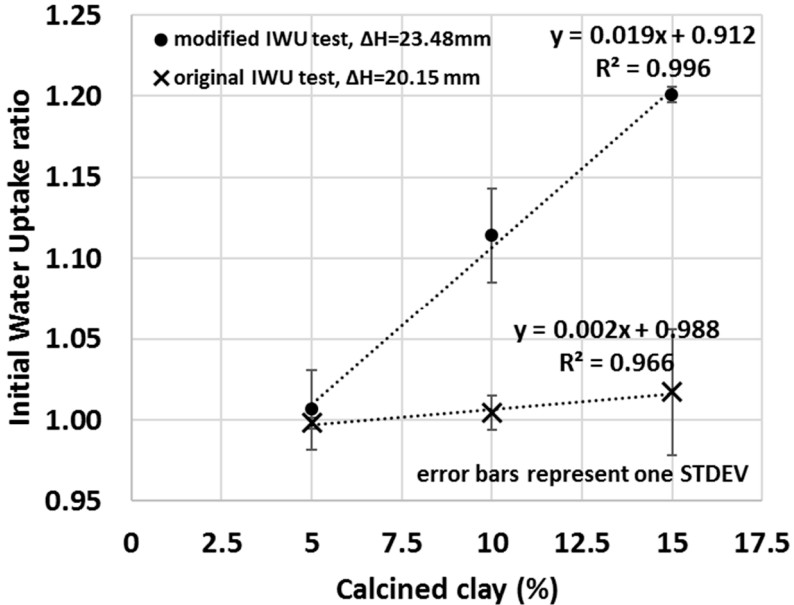

**Figure 9.** The impact of using a greater capillary height differential between the water level and the sample (X symbols—original height difference; • symbols—increased capillary height differential) on the IWU ratio.

Reduced workability of the calcined clay–OPC blends compared to that of OPC only is often considered a result of the shorter setting times of the blends [59,60]. However, to achieve normal consistency, which is a criterion of the setting time test (ASTM C191, [61]), it is also clear that calcined clay–OPC blends require more water than that of OPC [62,63]. However, calcined clay blends at normal consistency tend to have similar setting times compared to those of OPC-only systems [64,65]. The results suggest that the workability of calcined clay–OPC (or other SCM–OPC) blends is influenced by the initial interaction between the SCM particles and water and that the change in setting time is more a consequence of this altered particle–water interaction (e.g., greater initial water uptake → less porous cementitious matrix → shorter apparent setting time) than the actual effect of the altered workability caused. This was one of the reasons for developing the IWU test.

It was hypothesized that the increased water uptake of calcined clay–OPC blends stems from the different structure and properties of calcined clay compared to those of OPC [40,44]. Upon calcination of kaolinitic clays such as those used herein, the raw clay is heated to ~800 °C to obtain metakaolin (i.e., calcined clay). During the heat treatment, the clay goes through dehydration and dehydroxylation and undergoes structural re-organization in the octahedral sheet (aluminium takes on a 5-fold rather than 6-fold coordination) forming metakaolin [39,66]. It is well known that kaolinites release high fractions of hydroxyl groups (as $H_2O$) upon heat-treatment [29,30,67,68]. The resulting coordination under-saturation of aluminium atoms in the calcined clay leads to a high

affinity to water and ultimately a high initial water uptake of metakaolin upon rehydration. Consequently, the initial wetting of calcined clay involves strongly bound water. The strong competition for water by calcined clays is quite possibly further enhanced by the spontaneous wetting of metakaolin that releases heat. This expelled heat changes the permittivity, viscosity and surface tension of water and thus alters its surface wetting properties. Furthermore, Jaskulski et al. (2020) [28] discussed that the layered structure of 1:1 clays such as metakaolin also has an important role in the observed higher water uptake. Such layered assembly can be seen in the SEM image of calcined clay used in this study (Figure 10).

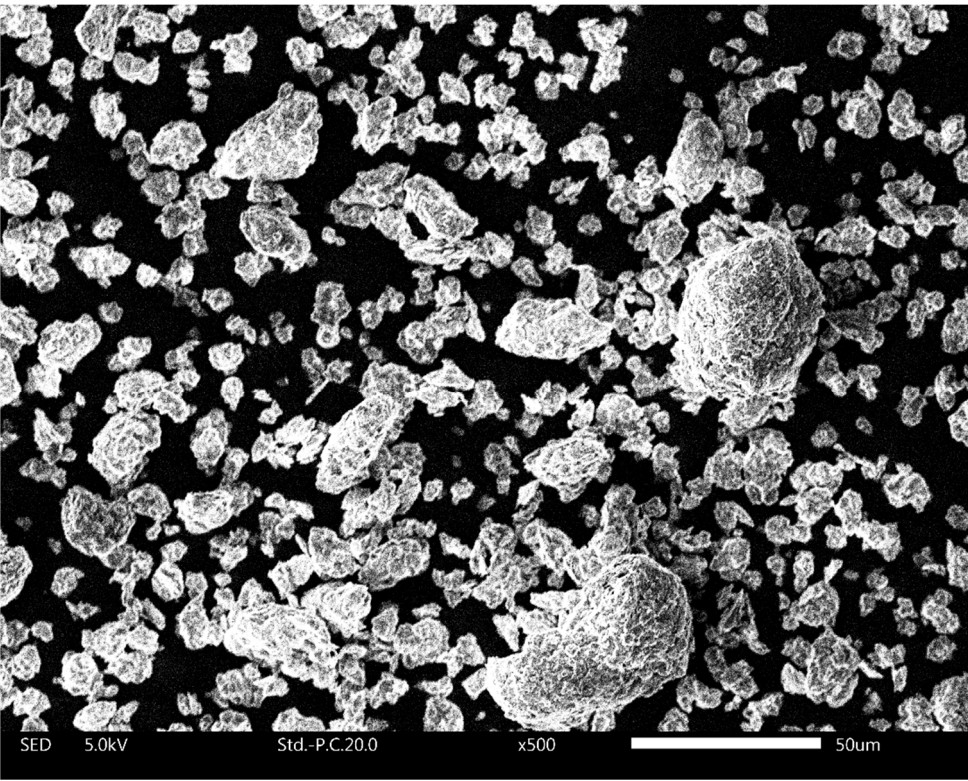

**Figure 10.** Scanning electron microscopy image of calcined clay particles used in this study.

By plotting the IWU ratios against either the mini slump values (Figure 11a) or the mini bleed values (Figure 11b), linear correlations can be drawn. The mini slump values, for example, gave a linear regression coefficient of $R^2 = 0.951$ and the mini bleed values gave $R^2 = 0.981$ with the IWU ratios.

Based on the measured IWU of the calcined clay–OPC blends, it is clear that the observed trends in the slump and bleed values were dominated by the initial water uptake by the SCM. For instance, if a larger portion of OPC was replaced with calcined clay, then the IWU was greater for the cementitious blend (i.e., less free water was present in the system), and consequently, less water was able to bleed out (Figure 7). In addition, less water is available to act as a "lubricant" and thus decreases the observed slump (Figure 5). Conversely, lower additions of calcined clay returned smaller IWU values, which ultimately led to greater bleed quantities and larger slump areas.

In a similar fashion, the measured mini bleed and mini slump values of limestone, fly ash and LC3 blends can be explained, although some other factors discussed below may also have impacted or even dominated the measured fresh properties. Fly ash addition, for example, exhibited a less pronounced effect on IWU compared to that of calcined clay (i.e., while negative, the absolute value of the slope of the linear regression for fly ash in Figure 8 is smaller than that of the calcined clay), and the linearity seemed to be valid for very low SCM percentages as well (note 0%, 15% and 30% fly ash additions were actually

measured and 0.999 $R^2$ value was found). The mini slump values for fly ash blends were larger than those of OPC which can be explained by a lower affinity of fly ash to water. Given that fly ash addition resulted in IWU ratios lower than 1, OPC has a greater affinity to water and if some of the OPC is replaced with fly ash then less water is taken up by the blend. Thus, in a fly ash–OPC blend, more water remains between the grains and particles as free water compared to that of an OPC–only system and this explains the measured greater mini slump values of fly ash blends compared to that of OPC.

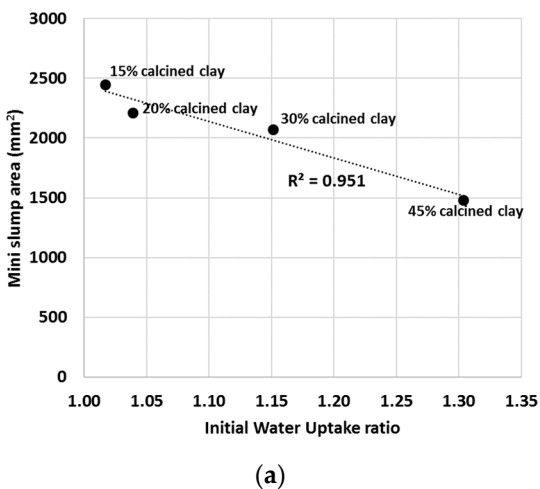
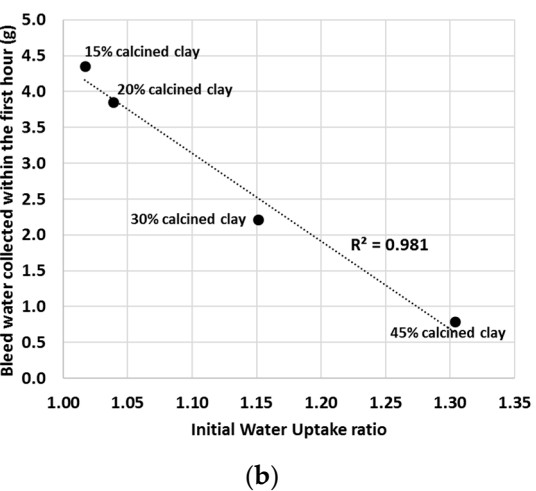

(**a**)             (**b**)

**Figure 11.** Linear regression between the mini slump values (**a**) or mini bleed values (**b**) and the IWU ratios of calcined clay blends. (Note that the linear relationship is obtained at calcined clay percentages between 15% and 45%.).

The greater bleed values of fly ash–OPC blends measured supported these observations. However, the blend obtained for the 30% fly ash replacement barely bled more water than that of the OPC-only and no linear trend was found between the amounts of bleed water and percentages of fly ash. These results suggest that parameters other than IWU may influence the resulting bleed of fly ash blends. For example, it is well known that the bleed water is the free water in the pores/channels of freshly mixed cementitious composites that cumulates at the surface during the consolidation of the cementitious matrix with time [69]. Thus, it is likely that the unexpectedly lower bleed value of the 30% fly ash–OPC blend is the result of the combination of lower water uptake (i.e., more free water and less water bound by the particles) on one hand, but restrained consolidation of the composite on the other hand (two opposite effects). The combination of these effects then ultimately resulted in a slightly greater bleed of the 30% fly ash–OPC blend compared to that of the OPC cement paste. It is therefore hypothesized that in the case of the 15% fly ash–OPC blend, the diminished IWU ratio (i.e., lower water uptake) was the dominating effect which resulted in a greater bleed value (because of more free water) and greater slump (greater lubrication effect due to the presence of more free water). Such behaviour of greater amounts (e.g., 30%) of fly ash in blends has not been previously reported and thus requires further study. However, the diminished heat of hydration due to the reduced amount of OPC in the 30% fly ash blend might explain the unexpected behaviour.

The mini slump and mini bleed values of the 15% limestone–OPC blend were in agreement with the obtained IWU ratio. The IWU test (Figure 8) indicated that limestone addition decreased the proclivity of the blend to absorb water and resulted in more water being located the pores, thus the measured greater slump (Figure 5) and detected more bleed water (Figure 7) is not surprising. However, the measured bleed water quantity was probably influenced by a slightly reduced compaction property of the cementitious matrix as well because the measured IWU ratio would indicate a greater bleed value of the 15% limestone–OPC blend than what was actually measured. Mini slump, on the other hand, showed very strong agreement with the measured IWU in terms of relative performance

compared with that of OPC (e.g., mini slump of OPC was ~73.5% of the mini slump area of the 15% limestone blend and the IWU of the blend was ~80.8% of the IWU of OPC).

As discussed in Section 3.3, the measured mini slump of the LC3 blend was somewhat surprising because even though calcined clay addition lowers the mini slump value and limestone increases the mini slump value, in the end, LC3 had a lower mini slump than that of the 30% calcined clay–OPC blend (LC3 contained 30% calcined clay and 15% limestone). However, the IWU ratio reveals (Figure 8) that there must have been an interaction between the calcined clay and limestone that increased the observed IWU of LC3 ($IWU_{LC3}$ = 1.1946 vs. $IWU_{30\%calcined\ clay}$ = 1.1511) and hence lowers its mini slump (Figure 5) (and bleed (Figure 7)). This interaction is maybe the early formation of the aforementioned carboaluminate hydrates that certainly increase the water demand of the system. Importantly, we emphasize that the combination of limestone and calcined clay is still highly beneficial, not just because of the superior hardened and durability properties [12,20], but also because it resulted in a much lower IWU ratio (and correspondingly, a greater mini slump) than that of the same SCM percentage (45%) of calcined clay only.

The overall higher IWU ratios of LC3 and calcined clay blends might play an important role in the hydration reactions and strength development of SCM–OPC blends. In this consideration, in a calcined clay–OPC blend, some of the water is initially held by the calcined clay. However, as the easily accessible water (not held by the calcined clay portion of the blend) is "consumed" by the OPC (especially by $C_3S$ in OPC) particle affinities for water change, between hydrating OPC and calcined clay. In other words, the calcined clay might act as a "buffer" that secures a steady but somewhat slower water supply for the hydration reactions of OPC. As known from chemistry, a slower crystal growth results in larger and more homogeneous crystals that could be beneficial in the final strength development of the cementitious composite. This consideration would explain the slow initial strength gain but greater final strength of LC3, for example [13]. The authors were aware that the hydration reaction of LC3 and calcined clay blends are quite complicated where pozzolanic reactions, monocarboaluminate formation and calcium alumino silicate hydrates (C–A–S–H) formation, just to mention a few reactions, take place and make the already complex hydration chemistry of OPC even more complex. Nonetheless, the role of initial water uptake is a phenomenon that is seldom considered when the hydration of calcined clay bends is discussed and hence the authors wanted to point out that it might play an important role since it exerts its impacts at an earlier stage.

Nevertheless, based on the results presented in this Section, it is very clear that IWU has a strong influence on the observed fresh properties of a cementitious composite and can be used to explain the measured slump or bleed. In real-world applications, the authors envision the IWU test being useful in optimizing compositions of cementitious blends and resulting mix designs for various SCMs combined in complex tertiary or quaternary systems.

### 3.5. Using IWU for Estimating Water Demand of SCM–OPC Mortars

To test the usefulness of the proposed IWU test, a series of mortar mixes were prepared and their slump values were measured. Initially, OPC-only mortar mixes were made with arbitrarily selected water-to-binder ratios of 0.36, 0.42, 0.48 and 0.51. The water-to-binder ratio range was considered appropriate for most applications. Then, in a second series of mixes, the water demands of 20% calcined clay–80% OPC blends were established using the previously obtained IWU numbers. In this instance, the water-to-binder ratios of the 20% calcined clay blends were calculated by multiplying the water-to-binder values used in the OPC-only mixes with the IWU value measured for 20% calcined clay–80% OPC (in Figure 8, this value is 1.039, and thus the water-to-binder ratios were 0.37, 0.44, 0.50 and 0.53 for the 20% calcined clay mortars). The intention was to obtain the same slump values what were measured for the OPC-only mortar mixes.

The same approach was applied to another series of mortar mixes with 30% calcined clay–70% OPC. The IWU value was 1.151 for the 30% calcined clay blend and thus this

value was used to establish the required water-to-binder ratios (0.41, 0.48, 0.55 and 0.59). As depicted in Figure 12, all cementitious materials (pure OPC and calcined clay blends) showed linear trends between the measured slump values and water-to-binder ratios used. Importantly, however, the IWU values determined from the initial testing, correctly estimated the required water-to-binder ratios for calcined clay blends only at higher water-to-binder ratios (top and second lines in Figure 13).

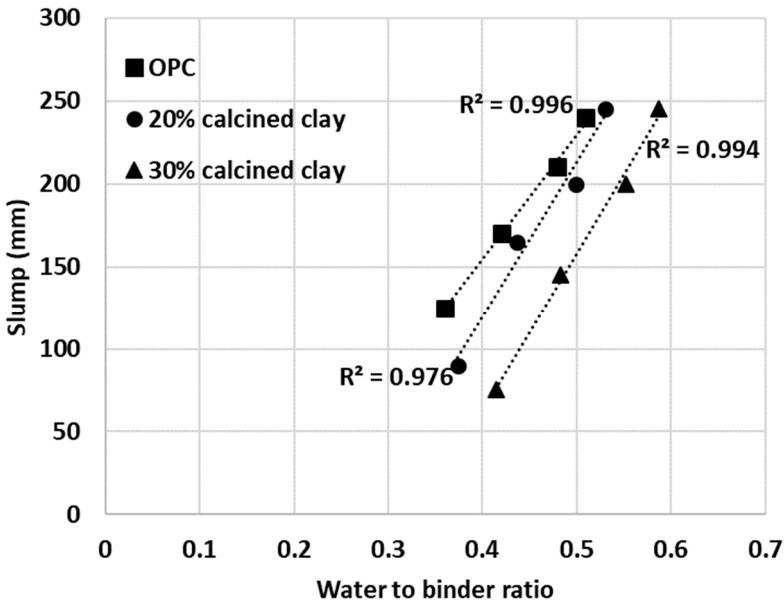

**Figure 12.** Measured slump values of mortar mixes with various water-to-binder ratios.

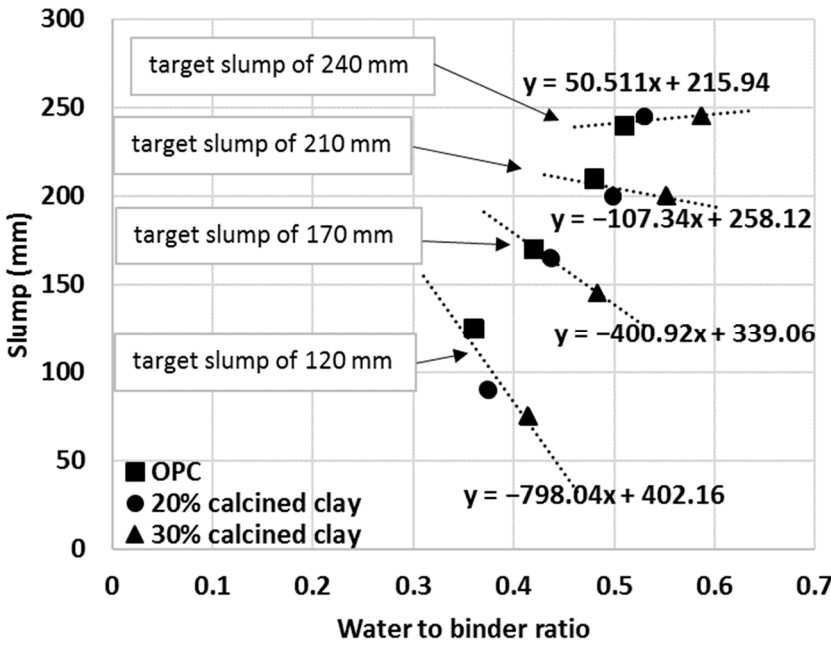

**Figure 13.** Linear relationships to judge the goodness of IWU values for predicting the water demand of various mortar mixes.

To further investigate the usefulness of the IWU values for calculating the required extra amount of water for blends having similar workability as the reference OPC, the targeted slump values were grouped based upon the initial water-to-binder ratios of OPC. For example, the first data point of the top line in Figure 13 represents the OPC mortar with a 0.36 water-to-binder ratio and 240 mm measured slump. The middle point in the

same line represents the water-to-binder ratio required (w:b = 0.37) for a 20% calcined clay blend to achieve the target same slump value (240 mm) calculated using the IWU value determined for the given blend:

$$\text{w:b}_{20\%\text{-calcined clay}} = \text{w:b}_{\text{OPC}} \times \text{IWU}_{20\%\text{-calcined clay}} = 0.36 \times 1.039 = 0.37.$$

Likewise, the third data point for that line represents the w:b ratio required to achieve the same slump value for a 30% calcined clay mortar:

$$\text{w:b}_{30\%\text{-calcined clay}} = \text{w:b}_{\text{OPC}} \times \text{IWU}_{30\%\text{-calcined clay}} = 0.36 \times 1.151 = 0.41.$$

Each line in Figure 13 represents different targeted slump values (from bottom to top: 120 mm, 170 mm, 210 mm and 240 mm).

In Figure 13, a horizontal (or close to horizontal) line represents an accurate estimate of the water-to-binder ratio with the IWU value to obtain the same slump value as measured for the given OPC mortar. In other words, the correctness of the IWU value for a particular cementitious blend can be evaluated from the resulting slope of the (linear) relationships in Figure 13. One can see that the obtained IWU values give a good estimation of extra mixing water required for the higher water-to-binder ratio (and higher slump) region. For these cases, the slopes of the linear trend lines are lower values (e.g., for the second line from the top the slope is $-107.34$ and the measured slump values are $\text{Slump}_{\text{OPC}} = 200$ mm, $\text{Slump}_{20\%\text{-calcined clay}} = 210$ mm and $\text{Slump}_{30\%\text{-calcined clay}} = 210$ mm—note that the accuracy of the standard slump test is 10 mm.). It is interesting that even for a higher slope value (e.g., third line from the top: Slope = $-400.92$), the estimation of the required water-to-binder ratio is not bad (for this instance, the measured slump values are the following: $\text{Slump}_{\text{OPC}} = 170$ mm, $\text{Slump}_{20\%\text{-calcined clay}} = 165$ mm and $\text{Slump}_{30\%\text{-calcined clay}} = 145$ mm). It is also obvious that as the water-to-binder ratio lowers the prediction for required water-to-binder ratio is underestimated (and may indicate the need for an appropriate water reducer admixture). Such nature of the estimation might be related to the way the IWU values are obtained. During the IWU test, there is plentiful water present and this indicates that the IWU test accurately predicts the required water-to-binder ratio for various SCM–OPC blends if there is enough free water in the system (e.g., if the water is enough to cover all surfaces and pores of the aggregates and the remaining amount of water available for the cementitious materials is sufficient). This finding suggests that the proposed IWU test is particularly useful for high slump and self-compacting concrete (SCC) type of applications with SCMs. For low water-to-binder applications, the solution might be to optimize the IWU test as described in Section 3.4, by increasing the capillary height of the experimental setup. However, low water content scenarios need to be investigated in the future to verify this.

### 3.6. Justification of the IWU Test Parameters

In this section, an insight is given toward what considerations and optimizations led to the proposed method for measuring the initial water uptake of cementitious materials. During the development of the experimental setup, there was an intention to use inexpensive equipment that is easy to operate and has the flexibility to study a wide variety of materials. Once the final design of the IWU test was established, a few test parameters were considered critical: the mass of the sample, the duration of the test and the capillary height differential established between the water level and sample. The establishment of the first property was by a trial and error found that 5 g of cementitious material was the optimum amount to obtain a thin, continuous and uniform layer of cementitious material within the given test area (75 mm-diameter circle, Figure 1). Although 4 g, 5 g and 6 g sample sizes of OPC showed similar gravimetric water contents after 1 min hydration ($\sim 47\% \pm 0.7\%$), a greater sample size of 6 g resulted in a double standard deviation of the measured gravimetric water ($\text{STDEV}_{5g} = 0.5011$, $\text{STDEV}_{6g} = 1.0822$) and a smaller sample

size of 4 g was just sufficient to cover the whole area of the D = 75 mm circle. Consequently, a 5 g sample size was considered ideal.

The optimal duration of the test required more attention and at first, the amount of water migrated into 5 g OPC after various hydration times was measured (Figure 14). Since one of the intentions of the proposed IWU method was to gain information on the early interactions between water and cementitious materials, very short test durations (e.g., 15 s, 22 s, 30 s and 45 s) were also tested. As shown in Figure 14, increasing the test duration of OPC hydration returned in a saturation type of curve where initially, the gravimetric water content (GWC) of the OPC sharply increased with time, then the changes dramatically slowed down and became essentially static at or above 1 min hydration time. These experiments resulted in the conclusion that the optimal test time of the IWU test is 1 min.

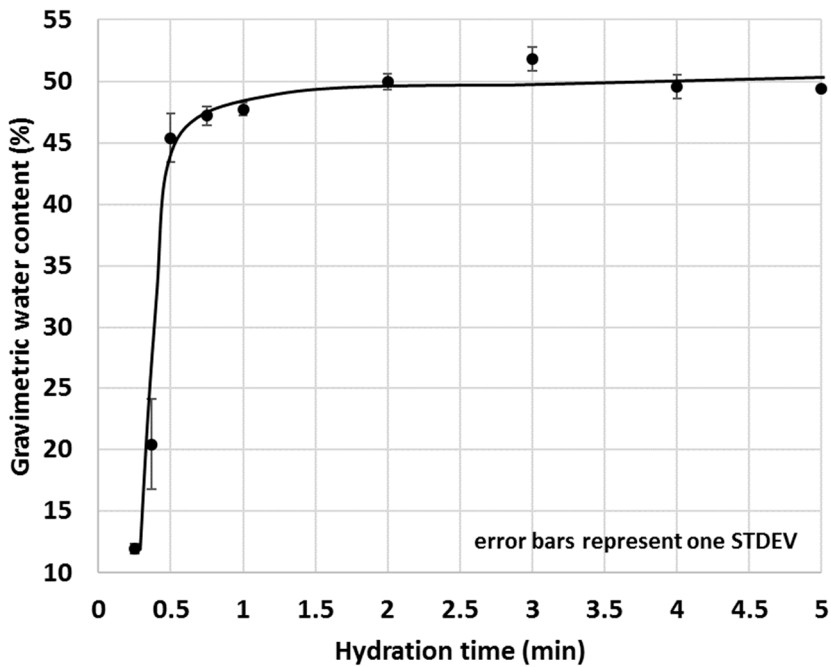

**Figure 14.** Gravimetric water content of 5 g OPC after various hydration times.

The selected 1 min test duration was then trialled in an actual test scenario where the initial water uptake of a selected SCM–OPC blend (i.e., the GWC of the SCM–OPC blend related to the GWC of pure OPC) was measured with time. A 30% calcined clay–OPC was chosen for the given purpose because this blend gave a medium (i.e., neither too high nor too low) IWU ratio in previous tests (Figure 8). Based on the studies on OPC only (Figure 14), water uptake for OPC very rapidly increase (high growth section of the saturation curve) within the first 30 s and hence for the experiments where the IWU of 30% calcined clay–OPC blend was studied, 30 s was chosen as the shortest test duration. As indicated in Figure 13, the IWU of the 30% calcined clay–OPC showed a very similar trend to what was observed for the GWC of OPC (Figure 14); namely it reached its plateau after 1 min. Thus, a 1 min test duration was considered appropriate for testing the IWU. It is noted that the proposed 1 min test duration might be too short for samples having larger particles in which water uptake is significantly slower, and for these circumstances, a few trial tests are recommended in order to establish the appropriate test duration. However, the authors are of the point of view that the particles' size distribution of the OPC used in this study is typical in many countries of the world.

As a final remark on the duration of the IWU test, it is worth noting that the measured standard deviation values within the high growth section of the above curves (Figures 14 and 15) were significantly greater compared with those of the sections of the curve after (or before) the rapid growth. This is because the water uptake before or after

the high growth section was almost static (either negligible amounts of water is taken up or a larger amount of water is already taken up and the further water uptake is limited). Thus, the greater spreads within the growth section of the curves are due to the dynamic (active) water uptake behaviour of the cementitious materials where even a 1 s difference of test duration can result in a significant change of measured GWC.

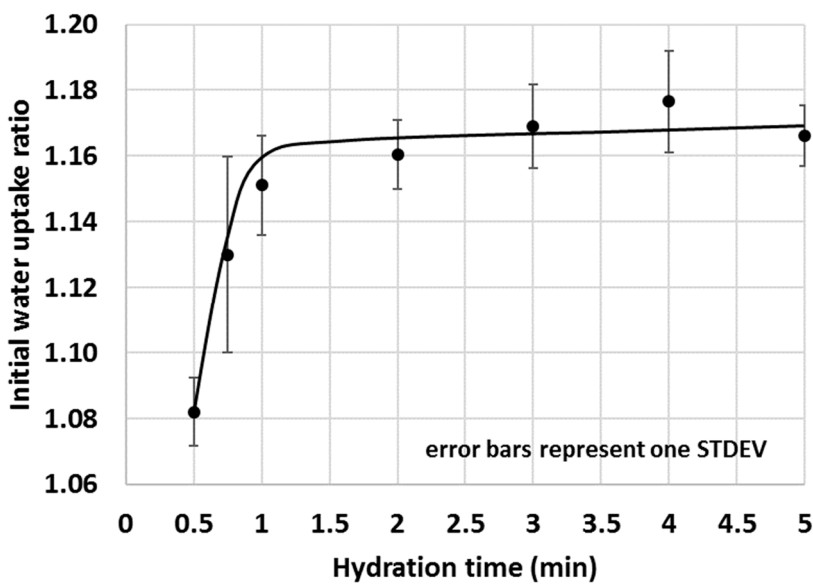

**Figure 15.** Initial water uptake of 30% calcined clay–OPC blend with time.

The capillary height differential between the water surface and the sample was selected so that most of the SCMs tested herein gave a relatively low standard deviation of measured gravimetric water content (i.e., the applied height difference with the selected test duration gave a water content that laid after the "high-grow" section of the hydration curve). As shown in the previous section (Figure 9), the height difference can be easily optimized for various types and amounts of SCMs.

### 3.7. Other IWU-Related Considerations

Throughout this study, there was also an intention to work with similar particles (in terms of size and shape) of various SCMs; thus, some of the materials were ball-milled to obtain particles owning similar properties to those of OPC (see Section 3.1). This criterion was considered important because we wanted to make sure that any difference in the measured IWU ratios did not originate from the different physical features of the SCM particles blended together with OPC. This condition is not considered important for the general user because the proposed IWU method intends to measure differences in IWU due to particle size/shape differences as well. However, during method development, the discrepancies in particles' size and shape were avoided to make the tested systems and the understanding of the IWU of the various materials less complex.

For a single test (30% calcined clay–OPC blend) the effect of particle size was investigated. In this case, the calcined clay component of the blend was further ground manually for 10 min by using a mortar and pestle. Non-surprisingly, the blend with the finely ground calcined clay picked up more water within the test duration and resulted in a ~4% increase in the measured IWU ratio (1.1982 vs. 1.1511). Thus, the particle size could have an impact on the initial water uptake of a cementitious material and validates that the proposed method can capture small differences in water dynamics due to alterations in particle size distribution (in this instance, all other parameters of the blends were the same).

Finally, with a high number of tests, it was observed that ambient properties such as temperature and relative humidity (RH) had an impact on the water uptake behaviour of various cementitious materials. Firstly, the effect of ambient temperature on the gravimetric

water content of OPC after 1 min test time was evaluated (Figure 16) and an increasing trend of GWC with temperature was observed. RH impacted the measured GWC of OPC; and on the contrary, a decreasing trend of GWC with growing RH was observed (Figure 17). It was hypothesized that the two investigated parameters were not independent from each other, and to check this, the ambient temperature was plotted against RH. As anticipated, in general, the RH decreased with ambient temperature (Figure 18).

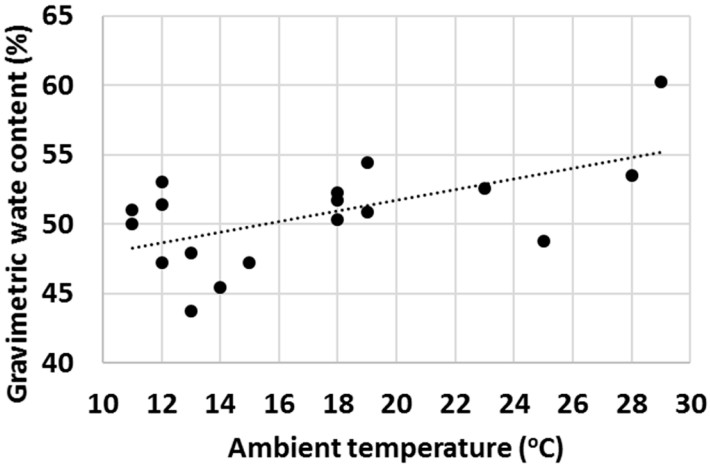

**Figure 16.** Gravimetric water content of OPC after 1 min hydration at various ambient temperatures.

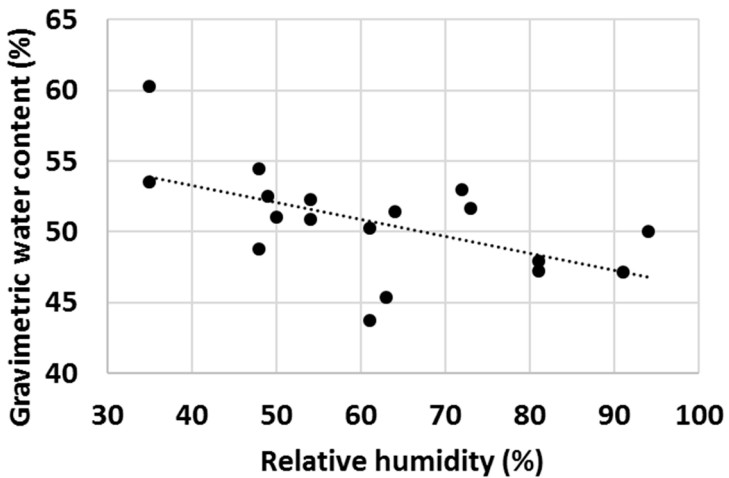

**Figure 17.** Gravimetric water content of OPC after 1 min hydration at various levels of relative humidity.

The authors are of the point of view that the actual differences in the measured GWCs of OPC were predominantly due to changes in the properties of water with temperature (water was stored in large container prior the water uptake test and hence had the same temperature as the ambient temperature). For example, the density, viscosity, surface tension, vapour pressure and thermal conductivity are all parameters that change significantly within the temperature range displayed in Figures 16–18 [70] and it is believed that a higher temperature led to overall properties of water which ultimately improved the water uptake. Viscosity is one parameter that is likely to have a substantial impact on the measured GWCs (e.g., by changing the temperature from 10 °C to 30 °C, the viscosity of water changes from 1.308 mPa s to 0.7978 mPa s). However, the proposed method of using a reference material (OPC) and taking the ratio between the GWC of the reference material and the GWC of the tested sample eliminated the temperature (and other ambient parameters) dependency of the IWU test. For example, the results presented in Figure 15 showed no extreme outliers even though the data collection was conducted on different

days with extreme variations ($\pm 7\,^{\circ}$C) in ambient temperature. The same is not true for Figure 14, where only the GWC of OPC is plotted. In this instance, the samples needed to be collected on the same day (with same ambient parameters); otherwise, significant fluctuations in measured GWC ($\pm 5\%$) were seen.

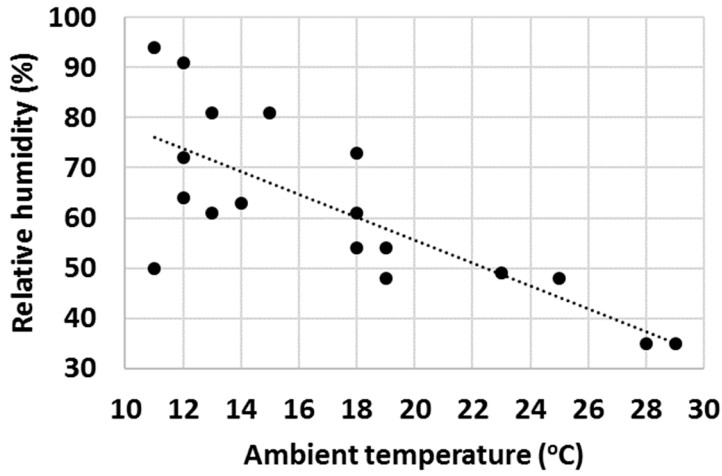

**Figure 18.** Changes in relative humidity with ambient temperature.

A final remark on ambient temperature: as displayed in Figures 16–18, the IWU test temperatures conducted in this study varied on a rather wide range (~10 °C–30 °C) because the tests were deliberately run in a room without temperature control. The aims of this experimental setup were multiple: (1) establishing the overall impact of ambient temperature on the measured GWC; (2) founding the relationship between ambient temperature and RH; (3) verifying that using a ratio of GWCs as the IWU value eliminates the temperature (and other external parameters) dependency of the test; and (4) develop a method that can be used in any laboratory in the world (we considered the 10 °C–30 °C temperature range wide enough to cover most laboratory conditions worldwide). Finally, the proposed method for measuring the IWU fulfils all these initial aims of the test program and is considered a cheap and easy method of gaining essential information on the early interactions between water and cementitious materials.

## 4. Summary and Conclusions

In this study, an initial water uptake (IWU) test was developed and evaluated for its ability to quantify the initial interactions between water and various cementitious materials. The proposed IWU test was desired because the currently existing tests for assessing fresh properties of cementitious composites lack capacity for providing information about the underlying phenomenon of the altered fresh properties upon using SCMs.

Ultimately, the IWU ratio is typically a number between 0.5 and 2 that reflects the affinity of a cementitious material (e.g., OPC or SCM–OPC blend) for water. An IWU ratio lower than 1 indicates that the given blend has a lower affinity to water compared to that of OPC and hence absorbs less water than OPC. Conversely, a ratio greater than 1 indicates enhanced water uptake and hence a greater water content after the initial water uptake reached a static state.

Based on measured standard deviations, trends with various types and amounts of SCMs and correlations with other tests, the IWU test is a very robust and highly reliable test and it can be used to explain the altered fresh properties of blended cements and to predict the water demand of cementitious mixes with SCMs. The IWU test was developed in a way (i.e., the incorporation of a reference material) that ambient conditions do not alter its performance and it can be used anywhere in the world without the control of the laboratory conditions.

Furthermore, the IWU test is a simple and low-cost test (using home-made apparatus), it can be used by inexperienced users and it quickly provides a quantitative result. The proposed method is intended to be an easily adaptable and highly valuable technique to all researchers and industry practitioners who routinely deal with SCMs and would like to evaluate the impacts of the selected SCMs on the fresh properties in advance or wish to understand the measured fresh properties or predict water demands.

Ultimately, the IWU test can be used to predict the altered water demand of mortar mixes incorporating SCMs to achieve a given slump value, with highly accurate predictions at higher water-to-binder ratios. We consider the IWU test to be useful for optimizing mix designs of SCM–OPC blends because it enables the differentiation and correlation of the water uptake of various SCMs with each other. The IWU test could prove useful in evaluating water relations in multicomponent (e.g., tertiary) cementitious blends.

In the future, (1) the effects of the IWU on the overall and specific qualities of mortars/concrete; (2) the impact of sample geometry on initial water uptake; and (3) the suitability of the IWU test for measuring the absorption of fine aggregates could be investigated. The latter one could help assess the potential for providing a surrogate method for the cumbersome water absorption test that is currently used worldwide.

**Supplementary Materials:** The following are available online at https://www.mdpi.com/article/10.3390/min11111185/s1, Table S1: Ball-mill program used in this study.

**Author Contributions:** Conceptualization, A.F. and W.P.G.; methodology, A.F. and W.P.G.; formal analysis, A.F. and C.G.; investigation, A.F., W.P.G. and C.G.; resources, A.F. and W.P.G.; writing—original draft preparation, A.F.; writing—review and editing, A.F., W.P.G., F.C.; visualization, A.F.; supervision, W.P.G. and F.C.; project administration, W.P.G.; funding acquisition, W.P.G. All authors have read and agreed to the published version of the manuscript.

**Funding:** This work was supported by the Australian Research Council's Special Research Initiatives scheme for PFAS remediation of soils, waters and debris, grant number SR180100009.

**Acknowledgments:** The authors would like to thank Jun Zhang and Anthony Antic for assisting in the particle size and shape measurements and Gyongyver Engloner for preparing some of the figures in this study. The authors also thank Claypro Australia Pty Ltd. for providing the clay material used herein.

**Conflicts of Interest:** The authors declare no conflict of interest.

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
