# Peer review of "A Porous Stone Technique to Measure the Initial Water Uptake by Supplementary Cementitious Materials"

_minerals, doi:10.3390/min11111185_

Round 1

Reviewer 1 Report

The paper is interesting, well-written and fills the identified gap in knowledge.

Just a minor remark:

Line 596 - please consider revising (is the verb missing?)

The main suggestion:

Could it be useful to provide a short note of explanation concerning table 1 - line 98 - regarding the issue of mortar quality - for construction, and how the changes in OPC-SCM blends could possibly affect the quality of mortar?

The conclusions:

What could be the correlation between the initial water uptake (IWU), OPC-SCM blends, and mortar quality?

Maybe a new field of research could be suggested?

Reviewer 2 Report

This paper presents a series of information on the use of methods for assessing the level of water absorption (IWU) for various cementitious materials.
- The paper, in terms of content, could be of interest to the scientific community;
- The introduction needs to be substantially improved as it is necessary to analyze information presented in other scientific papers;
- the research methodology must be presented more clearly. It is necessary to justify the decision to choose a certain composition for the samples shown in Table 1. Perhaps the mixtures analyzed in the study should have been made so that not a single parameter is varied. Thus, it may have been more interesting to use a much more complex experimental design in order to obtain the best results;
- macroscopic images of the test pieces taken and also microscopic images showing the internal structure of the test materials should be presented. Thus, it would have been possible to make a correlation between the determined properties and the structure of the materials;
- discussions need to be more detailed in order to highlight the contributions of research;
- the conclusions should be more concrete and encompass the  practical applications of the results obtained.

Reviewer 3 Report

This manuscript proposed a method to measure the initial water uptake of cementitious powders, and discussed its influence on the flowability and bleeding of cementitious paste. This method is interesting, and it is useful to the applications in laboratory and onsite. This manuscript can be accepted after addressing the following issues:

The title is too broad. It is better to specify its influence to workability, or indicate it is a method.

Line 21: There is no hydration kinetics studied in the main text. The authors should delete the description about “and ongoing hydration kinetics”.

Line 89: The LC3 was designed to contain 50 wt% clinker, 30 wt% calcined clay, 15 wt% limestone and 5 wt% gypsum. However, according to Table 1, the mixture LC3 includes 55% cement, 30 wt% calcined clay, 15 wt% limestone. These two mixtures are different, since the OPC contains 3 wt% gypsum, 6 wt% limestone and 91 wt% clinker. For the LC3 in Table 1, the calcined clay is higher 30%, the limestone is higher than 6%, and the gypsum is lower than 5%. Maybe this is the reason why you obtained unexpected results in the following part. Similar comments to the mix 15% limestone. Besides, all the mixes with only different calcined clay content also have a certain content of limestone, due to the presence of limestone in the Portland cement. This should be mentioned.

Line 114: Please give more details about the testing method of initial water contents of powders.

Line 137: During the mini bleeding test up to 3 hours, how to make sure the water was not evaporated?

Line 249: In the proposed initial water uptake test, the volume of the samples with different SCMs is different, due to the difference of specific gravity. Does the volume difference exert an impact to the measured results?

Line 313: Figure 4, it is better to change the x-axis from linear coordinates to logarithmic coordinates, otherwise, you cannot tell the difference based on the linear coordinates.

Line 347: The slump test is generally correlated to the dynamic yield stress, rather than static yield stress, see publications (J.E. Wallevik, Relationship between the Bingham parameters and slump, Cement and Concrete Research 36(7) (2006) 1214-1221; N. Roussel, Correlation between Yield Stress and Slump: Comparison between Numerical Simulations and Concrete Rheometers Results, Materials and Structures 39(4) (2006) 501-509; D. Jiao, C. Shi, Q. Yuan, et al., Effect of constituents on rheological properties of fresh concrete-A review, Cement and Concrete Composites 83 (2017) 146-159). Please do not mislead readers.

Line 423: The description “Blends at normal consistency also tend to have similar if not longer setting times” is not clear.

Line 430: If you measure the XRD pattens of the raw materials, this statement will be more convincing.

Line 441: You need to mention that the linear relationship is obtained at calcined clay replacement from 15% to 45%.

All the explanations about the strange results of LC3 should be re-phrased, because of the wrong mixture proportion in Table 1.

Line 592: The selection of the sample mass and optimal duration should be introduced in the testing method of initial water uptake, since the reviewer is also confused why you choose 5 g and 1 s duration when reading the testing method part.

Some language errors should be addressed, for example, line 352, line 400, line 431, etc.

Round 2

Reviewer 2 Report

The authors revised their manuscript according to my suggestions. Thus the manuscript can be accepted for publication.